# Collagen Fibril Diameter Distribution of Sheep Anterior Cruciate Ligament

**DOI:** 10.3390/polym15030752

**Published:** 2023-02-01

**Authors:** Smail Smatov, Fariza Mukasheva, Cevat Erisken

**Affiliations:** Department of Chemical and Materials Engineering, Nazarbayev University, Astana 010000, Kazakhstan

**Keywords:** ACL, sheep, collagen fibrils, diameter distribution, PCL, tissue engineering, nanofibers, biomimicry, ligament construct, electrospinning

## Abstract

The anterior cruciate ligament (ACL) tissue is a soft tissue connecting the femur and tibia at the knee joint and demonstrates a limited capacity for self-regeneration due to its low vascularity. The currently available clinical procedures are unable to fully restore damaged ACL tissue, and tissue engineering can offer options with a potential of restoring the torn/ruptured ACL by using biomimetic constructs that are similar to native tissue in terms of structure, composition, and functions. However, a model substrate to understand how the ACL cells regenerate the injured tissue is still not available. In this study, it is hypothesized that the nanofiber-based model substrate with bimodal and unimodal fiber diameter distributions will mimic the diameter distribution of collagen fibrils seen in healthy and injured sheep ACL, respectively. The aims were to (i) create an ACL injury in a sheep ACL by applying extensional force to rupture the healthy ACL tissue, (ii) measure the collagen fibril diameter distributions of healthy and injured ACL, (iii) fabricate polycaprolactone (PCL) nanofiber-based model constructs using electrospinning with diameter distributions similar to healthy and injured ACL tissue, and (iv) measure mechanical properties of ACL tissue and PCL electrospun constructs. The results showed that the fiber diameter distributions of PCL electrospun constructs and those of the healthy and injured ACL tissues were similar. The novelty in this investigation is that the collagen fibril diameter distribution of healthy and injured sheep ACL tissues was reported for the first time. The study is significant because it aims to create a model construct to solve an important orthopedic-related clinical problem affecting millions of people globally. The model construct fabricated in this work is expected to have an important impact on ACL regeneration efforts.

## 1. Introduction

ACL-related injuries represent a significant portion of musculoskeletal-related injuries, with around 4 million annual cases worldwide [1,2]. Additionally, this number shows an increasing trend, especially among the professional sportsmen including football players, skiers, and athletes [3,4]. Prior reports demonstrate that ACL injuries can potentially cause osteoarthritis [1] and secondary complications such as structural changes in cartilage and meniscus because the ACL has a limited potential of healing itself due to its low cellularity and vascularity [5]. The gold standard for ACL repair involves reconstruction of the ruptured ACL with natural and synthetic grafts but only at the expense of added morbidity and pain. Synthetic grafts are plausible options because their structural and morphological properties can be tuned to meet the needs. However, these grafts lack a significant physiologic property of the native ACL tissue, such as the organization and diameter distribution of collagen fibrils. The objective of this study is, therefore, to create a nanofiber-based model to be later used as a construct for investigating ACL regeneration mechanism. 

ACL tissue consists of water (~65% by weight) and solid materials. The biochemical composition of solid structure (~75% *w*/*w*) is Col-I (85%), Col-III, -VI, -V, -XI, and -XIV. The other 25% of solids contains proteoglycans (<1%), elastins, and glycoproteins [6]. The collagen fibrils in the ACL are secreted by fibroblasts and fibrochondroblasts, with diameters in the range 35–75 nm [7]. 

Structural and morphological characterization of collagen in the ACL is of great importance because the mechanical properties of the ACL are mainly associated with the collagens [8]. The stress–strain diagram of ACL tissue under tension clearly demonstrates a toe region, linear elastic region, yielding behavior, and finally failure [9]. Therefore, in order to recapitulate these properties and achieve an optimal ACL repair/regeneration, a physiological range of fibril diameter seems essential.

The diameter distribution of ACL fibrils was previously shown to change from bimodal to unimodal distribution upon injury, with reduced mean diameter for rats and bovine; however, the fibril diameter for injured ACL is not reported for sheep (Table 1). 

Different biomaterials including PLGA (poly(lactic-co-glycolic acid)) [16,17,18,19], PCL [16,17,20,21,22,23,24,25,26], PUR (polyurethane) [21], and PEUUR2000 (poly(ester urethane urea) [20] were earlier tested for ligament as well as ligament-to-bone interface regeneration. In this study, PCL was selected as the material of construction for the model constructs because of its feasibility in terms of cost and ease of electrospinability. Additionally, PCL is approved by the FDA to be used as a biomaterial for medical devices. 

Considering the processing of the PCL, the method of electrospinning enables production of nanostructures with desired and controlled fiber organization [27], which can replicate collagen fibrils in ACL [13,15].

With a goal of fabricating a nanofiber-based model construct, here, we hypothesize that the constructs will have a fiber diameter distribution similar to that of healthy and injured sheep ACL tissue.

## 2. Materials and Methods

In this study, first, the knee joints were harvested and stored at −20 °C until characterization. The procedure is graphically described in Figure 1. After thawing, the knees underwent mechanical testing to create injured tissues. The ruptured ACLs and intact ACLs were extracted from sheep knee joints and prepared for TEM analysis for diameter distribution measurements. Finally, the PCL scaffolds were fabricated using electrospinning, and their mechanical properties and diameter distributions were measured.

### 2.1. Materials

The following materials were used in this study and were procured from Sigma-Aldrich: pyridine (#270970), 2.5% of glutaraldehyde solution (#G5882), phosphate buffer solution (#P5244), 1% osmium tetroxide (#75633), ethanol (#E7023), propylene oxide (#82320), epoxy embedding medium 812 substitute (#45345), epoxy embedding hardener DDSA (#45346), epoxy medium accelerator DMP 30 (#45348), epoxy embedding hardener MNA (#45347), acetic acid (#695092), polycaprolactone MW = 80000 g/mol (#440744), and formic acid (#1.10854). 

### 2.2. ACL Tissue Isolation

The sheep knee joints were obtained from a local slaughterhouse (*n* = 10), frozen at -20 °C, and thawed at the time of characterization. The ACL tissues were extracted carefully to keep the structure of the tissue undamaged and prepared for characterization. The procedure was explained in detail elsewhere [13,28].

### 2.3. Transmission Electron Microscopy

ACL specimens were extracted from the middle portion of the healthy tissues (*n* = 4) and from the point of rupture of injured ACL tissues (*n* = 4). They were then cut into smaller specimens with dimensions of 1 mm × 1 mm for preparation for transmission electron microscopy (TEM) imaging (Figure 1C). 

The procedure is described in our previous report [13]. Briefly, the specimens were fixed in a solution of 2.5% of glutaraldehyde (Figure 1D) and were placed in refrigerator for gradual cooling to 4 °C. The specimens were then rinsed with phosphate buffer solution and fixed with 1% osmium tetroxide. Next, the samples were rinsed with PBS solution two times for 10 min, followed by ethanol and propylene oxide treatment. The dehydrated specimens were saturated with propylene oxide and resins. Specimens were immersed in the resin and propylene oxide (1:1) at 37 °C for 2 h. The resin propylene oxide ratio was then increased to 3:1 for 2 h, followed by pure resin for 12 h. The embedded samples were polymerized for two days at 60 °C. The hardened samples were cut in thin films using a microtome (Boeckeler RMC Power Tome Ultramicrotome, Boeckeler Instruments, Inc., Tucson, AZ, USA). Lastly, a TEM (JEOL JEM-1400 Plus 120 kV TEM, JEOL USA, Inc., Peabody, MA, USA) was used for taking the micrographs of ACL sections (Figure 1F,G). Imaging provided about 5–10 sections for each sample. 

### 2.4. Measurements of Fibril Diameter

The diameters of collagen fibrils seen on the TEM images were measured using ImageJ (National Institute of Health, Bethesda, MD, USA) software. Ten horizontal lines with equal distance were drawn on the image, and the diameter of the fibrils intersecting the lines was measured. The same procedure was also used to evaluate the fiber diameters of PCL constructs using their Scanning Electron Microscope (SEM) images.

### 2.5. Fabrication of PCL Constructs 

Unimodal (unaligned) constructs were fabricated using a single concentration of PCL, while bimodal (aligned) constructs were fabricated by electrospinning two different concentrations of PCL. Unimodal distributions were obtained by dissolving 0.5 g of PCL in the solution of acetic acid and formic acid with 0.05 mL of pyridine. (Final volume is 5 mL containing 11% *w*/*v* PCL.) The PCL solutions for bimodal fiber distribution were prepared using 0.2 g and 0.375 g PCL (8% and 15% *w*/*v* PCL concentrations) in 0.015 mL and 0.05 mL of pyridine, respectively, with respective amounts of formic and acetic acids in a total volume 5 mL. These preparations were mixed for 2 h at 1500 rpm at 40 °C to make them homogeneous. 

Aligned nanofibers with bimodal distribution were prepared by co-electrospinning (in opposite directions) of 8% and 15% PCL concentrations. These solutions were placed into two opposite syringes and electrospun towards the drum rotating at 2000 rpm. The voltage was 9 kV, and the distance between needle and drum was 7 cm. The unaligned fibers were fabricated with 11%PCL at a flow rate of 0.03 mL/h and a voltage of 9 kV and collected on a stationary surface.

### 2.6. Mechanical Properties and Injured ACL Model 

The injured ACLs were created by using a uniaxial deformation (Tinius Olsen H25KS, Horsham, PA, USA). Specimens (*n* = 4/group; dimensions are given in Table 2) were pre-strained at 4 N and then loaded continuously at 5 mm/min until rupture. The ruptured ACLs were used to represent injured ACL. 

Identical tensile tests were performed for PCL constructs (*n* = 5/group; dimensions are given in Table 3 and Table 4) via uniaxial universal mechanical testing device (MTS Criterion Model 43, MTS Systems Co., Eden Prairie, MN, USA). 

### 2.7. SEM Characterization

The specimens were coated (Quotrum Q150T ES, UK) with 3–5 nm thickness of gold and observed using an SEM (JSM-IT200(LA), JEOL, Tokyo, Japan). Fiber diameters were measured using ImageJ (National Institute of Health, Bethesda, MD, USA) software.

### 2.8. Statistical Analysis

Comparison of (i) collagen diameter of healthy and injured ACL tissues, (ii) diameter of PCL fibers in the construct, and (iii) mechanical properties of aligned and unaligned constructs were all performed using unpaired Student’s *t*-test. Comparison of ACL tissue and electrospun constructs for mechanical properties was performed using one-way analysis of variance (ANOVA). The difference was considered significant for *p* < 0.05.

## 3. Results

### 3.1. ACL Fibril Diameter

Figure 2 shows the collagen fibril diameter (CFD) of healthy and injured ACL tissues together with respective TEM images (*n* = 4/group). Obviously, the injured ACL tissue exhibited a unimodal distribution, while healthy specimens had a bimodal distribution. In addition, the injured specimens exhibited an unaligned organization as indicated by the fibril cross-sections deviating from a circular shape. On the other hand, the healthy specimens had an organized structure as indicated by aligned collagen fibrils (Figure 2, A1–A4). 

Figure 3 shows the distributions of injured and healthy ACL fibrils when the individual distributions were combined to form an average fibril diameter distribution. The injured tissue had a unimodal distribution with a single peak at 101 ± 9.1 nm. The healthy ACL tissue demonstrated two peaks at 75.6 ± 8.5 nm and 157 ± 3.8 nm. The range narrowed from 24–291 nm to 36–230 nm. The average mean diameter decreased from 124.2 ± 16.1 nm to 86.2 ± 12.7 nm (*p* < 0.05; Figure 3C). Overall, the mean diameter of sheep ACL collagens was reduced, and the diameter distribution of collagen fibrils changed from a bimodal to a unimodal distribution after the injury. 

### 3.2. Fiber Diameter of PCL Constructs 

Diameter distributions of aligned and unaligned PCL constructs are depicted in Figure 4 in addition to respective SEM images (*n* = 5/group). Obviously, constructs with the aligned fibers demonstrated a bimodal distribution, whereas the constructs with unaligned (random) fiber structure showed a unimodal distribution. 

Figure 5 depicts the aggregated distribution for unaligned and aligned PCL constructs separately. Obviously, aligned constructs exhibited two peaks at 85 ± 10 nm and 155 ± 10 nm, while unaligned PCL constructs had a distribution with a single peak at 100 ± 15.1 nm. The mean fiber diameter decreased from 122.1 ± 5.9 nm to 101.5 ± 15.4 nm (*p* < 0.05). The range of PCL fiber diameter narrowed from 47–262 nm to 42–263 nm. Moreover, the mean weighted diameter went down from 131.2 nm to 110.8 nm (Figure 5C). 

Figure 6A,B show how well nanofiber constructs mimicked the diameters of native ACL tissue. There was no difference between the aligned PCL diameters and healthy ACL in terms of mean diameter. In addition, unaligned PCL fiber diameter and injured ACL diameter were found to be similar. However, it should be noted that the frequency of unaligned PCL fibers was higher than that of the injured ACL tissue (Figure 6B), meaning that more fibers with mean diameter were formed in the electrospinning process. This may be optimized by modifying the material and processing parameters of electrospinning following a parametric study.

### 3.3. Comparison of Biomechanical Properties of ACL Tissue and PCL Constructs

Figure 7A shows the stress–strain and load–extension behavior of the ACL tissue. The ACL tissue had a maximum stress and strain values of 23.1 ± 12.0 MPa and 42.9 ± 24.6%, respectively, and a modulus of 0.5 ± 0.2 MPa. The energy stored during straining (area under the curve during straining) was determined to be 600.3 ± 466.2 MPa. The ACL could be elongated by 42.9 ± 24.6 % with an ultimate load of 568.7 ± 213.6 N (Figure 7C).

The aligned PCL constructs representative of healthy tissue had ultimate stress and strain of 1.7 ± 0.11 MPa and 20.3 ± 6.5%, respectively. On the other hand, the ultimate stress and ultimate strain of the unaligned PCL constructs, representing the injured state of the ACL, were found to be 1.5 ± 0.3 MPa and 26.7 ± 1.8%, respectively.

The elastic moduli values of the aligned (representing the healthy ACL) and unaligned (representing the injured ACL) constructs were 0.1 ± 0.04 MPa and 0.11 ± 0.03 MPa, respectively. The strain energy density was determined to be 4.1 ± 0.9 MPa and 2.7 ± 0.5 MPa for aligned and unaligned constructs, respectively. When stretched to an elongation of 20.3 ± 6.5 and 26.7 ± 1.8%, the ultimate tensile load of aligned and unaligned constructs was 4.9 ± 0.49 N and 1.8 ± 0.2 N, respectively.

Comparison of the aligned/unaligned constructs and ACL tissue confirmed that the results for all parameters were significantly lower than those of the native ACL tissue (Figure 8).

## 4. Discussion

In this study, we compared the collagen fibril diameter and diameter distribution of healthy and injured sheep ACL. Furthermore, mechanical properties and the fiber diameter distribution of nanofiber-based electrospun PCL constructs were evaluated to investigate the similarity between the PCL constructs and the ACL tissue. The initial bimodal fibril diameter distribution of the native ACL tissue changed to unimodal upon injury, and the average diameter decreased after rupture. The diameter distribution of PCL fibers in the construct also exhibited a bimodal and unimodal distribution to both qualitatively and quantitatively mimic the characteristics of healthy and ruptured ACL, respectively.

The morphological characteristics of ACL collagen fibrils are related to its function. The collagen fibrils in ACL are organized in the form of bundles along the longitudinal direction, which optimizes its mechanical properties [14]. Based on our findings, a healthy ACL tissue possesses an aligned organization of collagen fibrils, which is deformed upon injury. Ruptured human ACLs and rat patellar tendon also showed a disorganized collagen fibril configuration [29,30]. A similar behavior was observed for injured bovine [13] and rat [10] ACL tissues.

It appears that change of collagen fibril dimeter distribution of ligament/tendon from bimodal to unimodal after rupture and a decreased mean diameter are common denominators across species. Bovine ACL [13], rabbit MCL [15], human ACL [15], mouse/rat PT [29], and rat ACL [10] all demonstrated morphological changes upon injury. A study on bovine ACL [31] reported peak diameters of ~100 and 250 nm for healthy tissue that consolidated into a single peak of 100 nm after rupture. Another study reported that the human ACL with an average diameter of 75 nm (range: 20–185 nm) decreased to 71 nm (range: 20–290) nm [32]. In another study, rat ACL exhibited peak diameters of ~60 and ~150 nm for healthy tissue, which then changed to a single peak with a diameter of 125 nm after rupture [10]. In the current research study, we evaluated collagen fibril diameter distribution of healthy and ruptured sheep ACL. Findings of this study demonstrated that the mean diameter of collagens went down significantly upon rupture, and the fibril diameter distribution changed from bimodal to unimodal. Rumian et al. [12] reported an average fibril diameter of 181 ± 14 nm and a bimodal distribution with peaks of 60 and 200 nm for a 3-year-old healthy ovine. The main difference between Rumian’s study and the current study is that the sheep model in this study is an immature animal (age: 1 year old). In this study, the mean collagen fibril diameters of sheep ACL in healthy and injured states were measured as 124 ± 16.1 nm and 86 ± 12.7 nm, respectively.

Earlier reports on ligaments and tendons provided the average fibril diameters and diameter ranges of healthy tissues only, with no attention to injured tissues [28,32,33,34] The studies performed by our group using bovine [13] and rat [10] ACL tissues reported the effect of injury on collagen fibril diameter distribution. The results reported by Beisbayeva et al. [13] on bovine ACL and results of this study for sheep ACL are comparable in that the mean fibril diameter peaks of the healthy bovine ACL tissue were 73.3 ± 11.5 nm and 213 ± 11.5 nm as compared to 75.6 ± 8.5 nm and 157.6 ± 3.8 seen in sheep ACL. A study performed on pediatric posterior collateral ligament (PCL), medial collateral ligament (MCL), and lateral collateral ligament (LCL) demonstrated similar mean collagen fibril diameters (MCL, 88.0 ± 26.0 nm; LCL, 93.3 ± 34.6 nm; PCL, 90.9 ± 34.0 nm); however, the fibril distribution profiles exhibited different modalities [35].

More recently, a study performed on the semitendinosus muscle tendon of humans reported that the mean fibril diameter in the immature group was significantly smaller than that of the young and adult tendon [36]. The distribution of the collagen fibrils changed from right-skewed in the immature group to flat in the adult group.

The PCL constructs with aligned and unaligned structures mimicked the collagen diameters of healthy and ruptured ACL both qualitatively and quantitatively. The average diameter of fibers of aligned PCL constructs and fibrils of healthy ACL were found as 122.1 ± 5.9 nm and 124 ± 16.1 nm, respectively. Similarly, the average diameter of fibers of unaligned PCL constructs (101.5 ± 15.4 nm) and collagen fibrils of ruptured ACL (86 ± 12.7 nm) were the same (*p* > 0.05).

The diameter and distribution of collagen fibrils of ACL were previously shown to have a direct impact on mechanical properties [15,32]. In native ACL tissue, the void between thicker fibrils is occupied by thinner fibrils to form a highly packed ECM with bimodally distributed collagens. This structural organization of fibrils generates stronger mechanical properties. When this highly packed structure is deformed, the ACL’s capacity to withstand physiologic loads is weakened, leaving the tissue mechanically inferior. Uniaxial tensile deformation of the ACL tissue showed that the sheep ACL tissue could be strained by 42.9 ± 24.6% against a load of 600.3 ± 466.2 N. In the current research study, we used an extensional crosshead speed of 5 mm/min based on our past experience [37].

The native ACL tissue exhibited improved properties as compared to both aligned and unaligned PCL constructs in terms of ultimate strain, ultimate stress, strain energy density, and modulus. The aligned PCL constructs outperformed their unaligned counterparts in maximum load. The unaligned construct, on the other hand, exhibited larger values of strain, which is consistent with previously reported data for aligned and unaligned PCL constructs [13,27].

There are some limitations in this research study. Biomechanical properties of injured ACL tissue could not be assessed due to a lack of appropriate apparatus. Such a measurement would enable us to compare the mechanical properties of unaligned PCL constructs and injured ACL tissue. Additionally, the biomechanical characteristics of ACL were investigated by applying a tensile load to the femur–ACL–tibia complex although it is known that deformations in the sagittal plane are the dominant reason of ACL injuries [38].

## 5. Conclusions

This research investigated the morphological properties of collagens of native sheep ACL before and after injury. It was demonstrated that the collagen fibril diameter distribution of ACL tissue changed from bimodal to unimodal after rupture, and the mean collagen fibril diameter decreased. Additionally, the bimodal and unimodal distributions of electrospun PCL constructs mimicked the healthy and injured ACL tissues both qualitatively and quantitatively. The findings of the current investigation are expected to have a significant contribution to ACL repair and regeneration efforts because the constructs fabricated and tested here could be used as model substrates to investigate cell behavior in the case of ACL injuries.

## Figures and Tables

**Figure 1 polymers-15-00752-f001:**
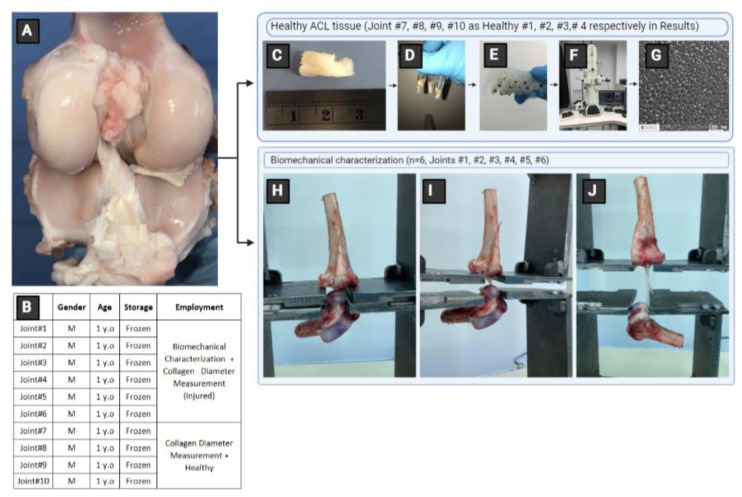
Procedure for ACL extraction and characterization. (**A**) Sheep knee joint, (**B**) characteristics of the harvested ACL tissues, (**C**–**G**) sample preparation and characterization using TEM, and (**H**–**J**) mechanical characterization (M, male).

**Figure 2 polymers-15-00752-f002:**
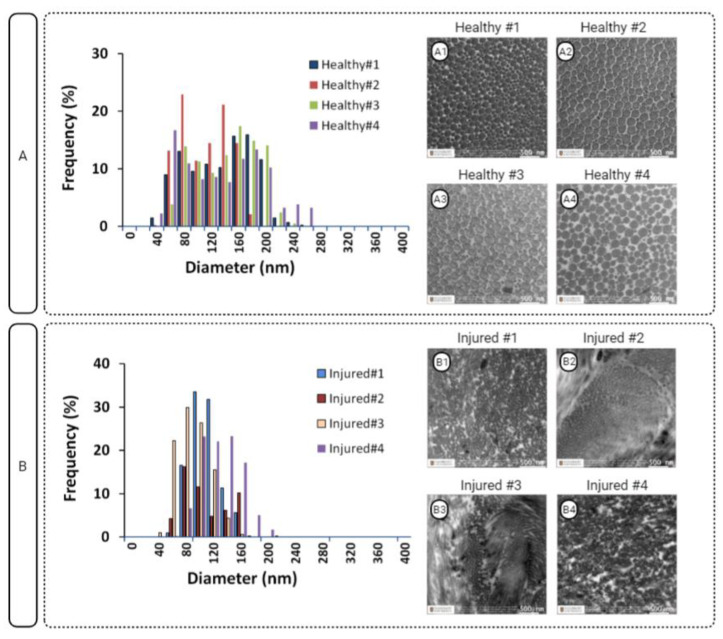
Diameter distribution of healthy (**A**) and injured (**B**) ACL fibrils and respective TEM images (A1–A4 and B1–B4).

**Figure 3 polymers-15-00752-f003:**
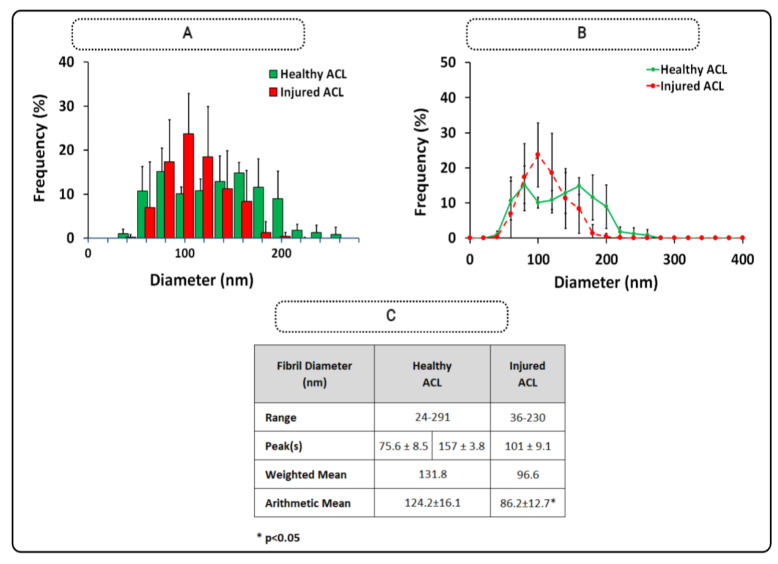
Diameter distributions of injured and healthy ACL collagens (**A**) and line graph (**B**) with descriptive statistics (**C**). * shows statistical difference at *p* < 0.05 (*n* = 4/group), and error bars represent standard deviation.

**Figure 4 polymers-15-00752-f004:**
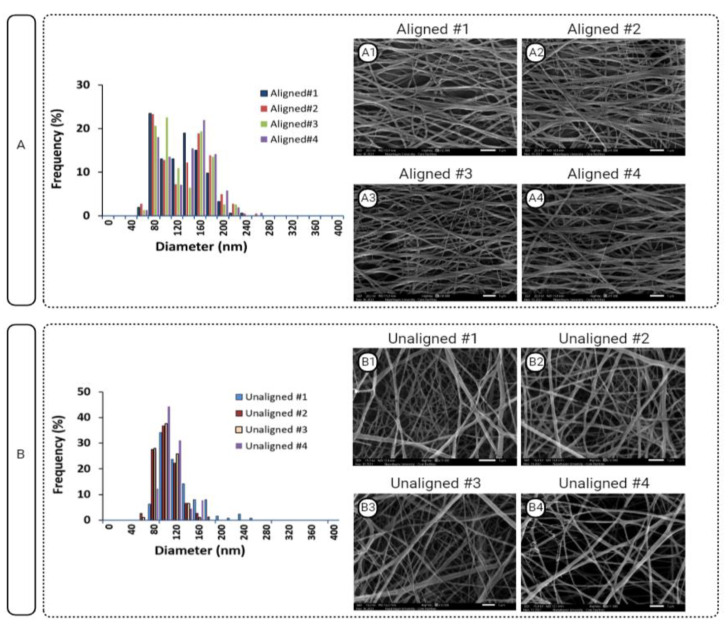
Distribution of fiber diameters of (**A**) aligned and (**B**) unaligned PCL constructs together with their corresponding (A1–4 and B1–4) SEM images. Scale bar = 1µm.

**Figure 5 polymers-15-00752-f005:**
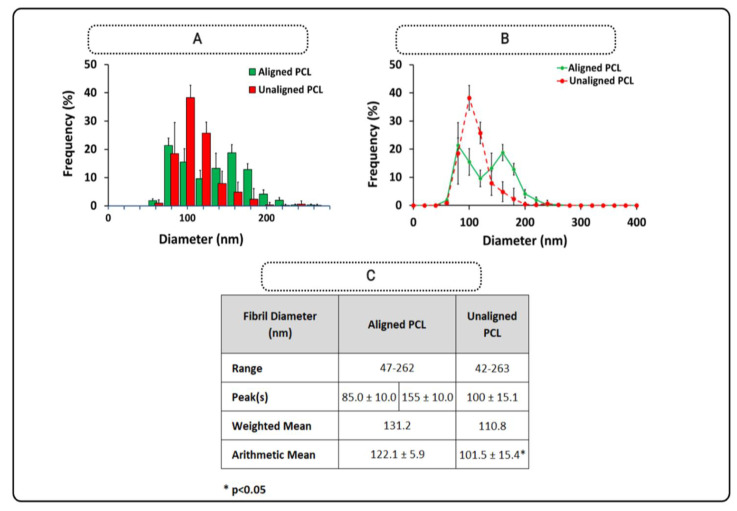
Aligned and unaligned PCL fiber diameter distributions. (**A**) Histogram, (**B**) line graph, and (**C**) descriptive statistics. * shows a significant difference at *p* < 0.05 (*n* = 4), and error bars represent standard deviation.

**Figure 6 polymers-15-00752-f006:**
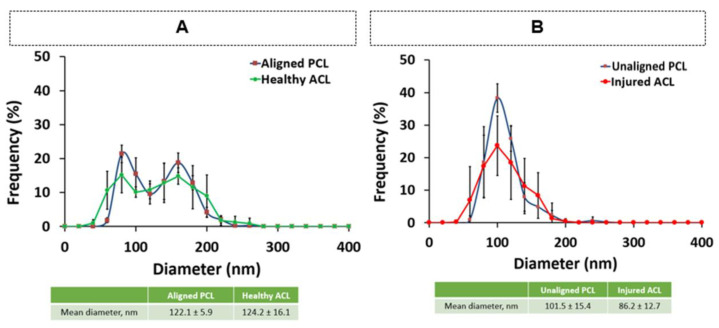
Comparison of diameter distributions of (**A**) aligned PCL vs. healthy ACL and (**B**) unaligned PCL vs. injured ACL. No significant difference was detected for either comparison at *p* < 0.05.

**Figure 7 polymers-15-00752-f007:**
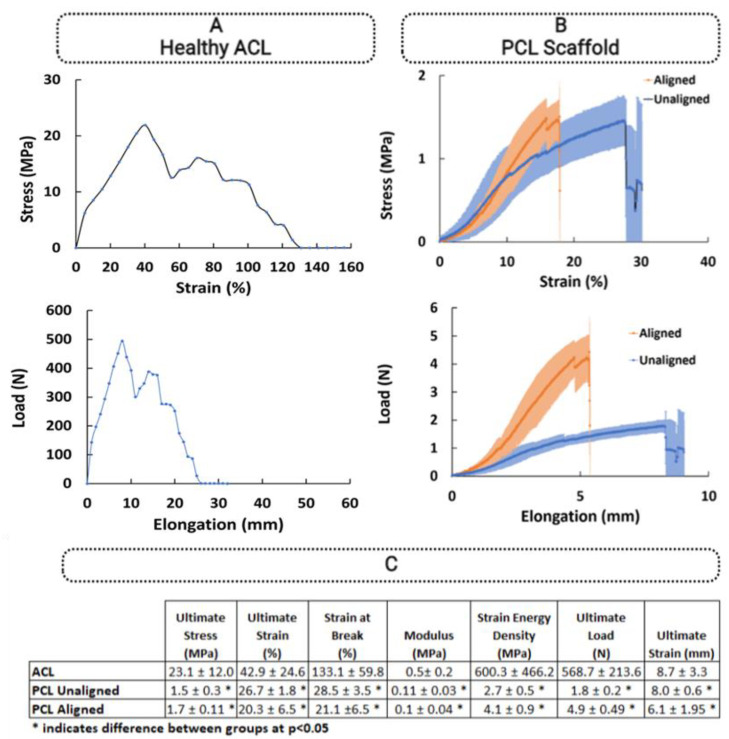
Mechanical properties of native ACL tissue and PCL constructs. (**A**) Healthy ACL tissue, (**B**) PCL constructs, and (**C**) descriptive statistics. * indicates significant difference at *p* < 0.05. Error bars represent standard deviation.

**Figure 8 polymers-15-00752-f008:**
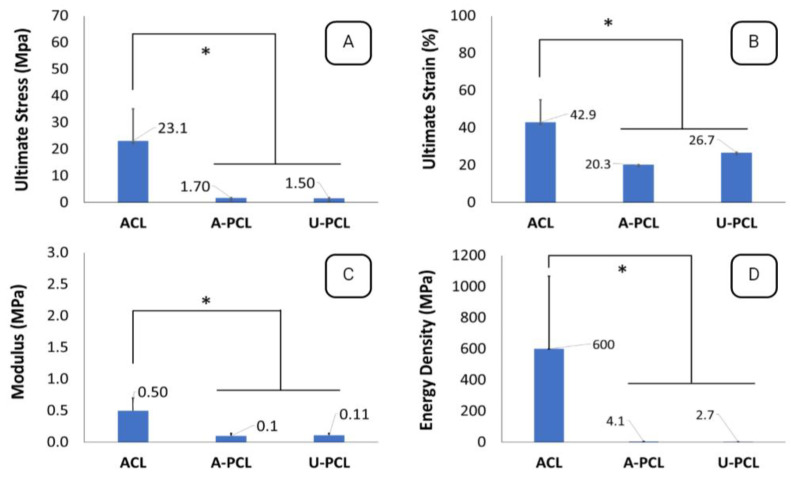
Comparison of ACL and electrospun constructs in terms of mechanical properties. (**A**) Ultimate Stress, (**B**) Ultimate Strain, (**C**) Modulus, and (**D**) Energy Density graphs determined at 5mm/min. * displays significant difference at *p* < 0.05. A-PCL, aligned PCL; U-PCL, unaligned PCL.

**Table 1 polymers-15-00752-t001:** Diameter distribution of collagen fibrils in previous reports.

Specie	Peak for Healthy Tissue (nm)		Range Healthy (nm)	Peak Value for Injured Tissue (nm)	Range Injured (nm)	Reference
	Smaller	Larger				
Rat ACL	~60	~150	20–260	~125	20–220	[10]
Rabbit ACL	~20	~250	10–320	NR	NR	[11]
Sheep ACL	~60	~200	20–300	NR	NR	[12]
Bovine ACL	~73	~210	20–320	~100	40–200	[13]
Human ACL	~50	~150	20–200	NR	NR	[14]
Human ACL	~75	NR	20–185	71	20–290	[15]

NR, not reported.

**Table 2 polymers-15-00752-t002:** Properties of ACL tissue for mechanical characterization.

	Thickness mm	Width mm	Area (mm^2^)	Length (mm)
Joint#1	3.59	6.05	21.74	21.30
Joint#2	3.77	4.51	16.99	15.67
Joint#3	4.69	6.11	28.64	16.09
Joint#4	4.30	5.34	22.99	25.14

Note: Joint#5 was determined to be an outlier and excluded.

**Table 3 polymers-15-00752-t003:** Properties of unaligned PCL scaffolds for mechanical characterization.

PCL 11%	Thickness mm	Width mm	Area (mm^2^)	Length mm
Scaffold#1	0.09	10.00	0.90	30.00
Scaffold#2	0.14	10.00	1.37	30.00
Scaffold#3	0.12	10.00	1.23	30.00
Scaffold#4	0.13	10.00	1.27	30.00
Scaffold#5	0.16	10.00	1.57	30.00

Note: Width of the scaffolds was determined by the cutting mold, and their length was determined by the gap of the mechanical testing device. These dimensions were not measured and are, therefore, constant.

**Table 4 polymers-15-00752-t004:** Properties of aligned PCL scaffolds for mechanical characterization.

PCL 8% + 15%	Thickness mm	Width mm	Area (mm^2^)	Length mm
Scaffold#1	0.23	10.00	2.33	30.00
Scaffold#2	0.29	10.00	2.93	30.00
Scaffold#3	0.29	10.00	2.93	30.00
Scaffold#4	0.30	10.00	2.97	30.00
Scaffold#5	0.26	10.00	2.60	30.00

Note: Width of the scaffolds was determined by the cutting mold, and their length was determined by the gap of the mechanical testing device. These dimensions were not measured and are, therefore, constant.

## Data Availability

Not applicable.

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
