# Peer review of "Collagen Fibril Diameter Distribution of Sheep Anterior Cruciate Ligament"

_polymers, 2023, doi:10.3390/polym15030752_

Round 1

Reviewer 1 Report

The submitted manuscript reports an interesting study about the formulation of PCL-based electrospun constructs mimicking the characteristics and the behavior of anterior cruciate ligament tissue.

Before recommending the manuscript for publication, the following issues need to be solved:

- The English language and style of the whole manuscript shold be improved to enhance the readability of the text. also the quality of the Figures needs to be improved.

- The reference list should be updted with some more recent papers on the same topic of the manuscript.

- Figure 2 could be removed, since a standard electrospinning apparatus is reported.

- In paragraph 2.6 details about the tested specimens should be added.

- I do not understand the presence of the error bars in figures 8A.

Reviewer 2 Report

Remarks to the author

This paper demonstrates interesting way of using PCL construct for Collagen fibril diameter distribution for sheep Anterior Cruciate Ligament. The significance of this work lies within the scope of well-established polymer-based research and innovation presented as the potential model in clinical system. Although adequate experimental results are included, the authors should further work on addressing some important points. Please refer to the below specific comments to work with:

Abstract

1.       Well written.

Introduction

1.       What is the diameter of the nanofiber? Why is it important to fabricate nano fibre in this work?

2.       What is the compatibility of the PCL for ligaments according to previous works?

Materials and Methods

1.       There is no description for Figure 1. I would recommend starting with brief description of Figure 1 in line 76.

Results

1.       I would suggest checking the term “unaligned” in the Figures properly.

2.       Does both graph in Figure 6 contain same information? If yes, I would suggest removing one.

3.       Is there any specific reason why aligned PCL has fluctuations?

4.       Is there any way to improve the performance of unaligned PCL for injured ACL?

Discussions

1.       Well written

Conclusions

1.       Satisfactory.

Round 2

Reviewer 1 Report

The manuscript can be accepted in the current form